# Diagnosis of Acute Leukemia by Multiparameter Flow Cytometry with the Assistance of Artificial Intelligence

**DOI:** 10.3390/diagnostics12040827

**Published:** 2022-03-28

**Authors:** Pengqiang Zhong, Mengzhi Hong, Huanyu He, Jiang Zhang, Yaoming Chen, Zhigang Wang, Peisong Chen, Juan Ouyang

**Affiliations:** 1Department of Laboratory Medicine, The First Affiliated Hospital, Sun Yat-Sen University, Guangzhou 510080, China; zhongpq@mail.sysu.edu.cn (P.Z.); hongmzh@mail2.sysu.edu.cn (M.H.); zhangj238@mail.sysu.edu.cn (J.Z.); chenyaom@mail3.sysu.edu.cn (Y.C.); 2Deepcyto LLC, 2304 Falcon Drive, West Linn, OR 97068, USA; hhy@deep-cyto.com (H.H.); alan@deep-cyto.com (Z.W.)

**Keywords:** artificial intelligence, acute leukemia, multiparameter flow cytometry

## Abstract

We developed an artificial intelligence (AI) model that evaluates the feasibility of AI-assisted multiparameter flow cytometry (MFC) diagnosis of acute leukemia. Two hundred acute leukemia patients and 94 patients with cytopenia(s) or hematocytosis were selected to study the AI application in MFC diagnosis of acute leukemia. The kappa test analyzed the consistency of the diagnostic results and the immunophenotype of acute leukemia. Bland–Altman and Pearson analyses evaluated the consistency and correlation of the abnormal cell proportion between the AI and manual methods. The AI analysis time for each case (83.72 ± 23.90 s, mean ± SD) was significantly shorter than the average time for manual analysis (15.64 ± 7.16 min, mean ± SD). The total consistency of diagnostic results was 0.976 (kappa (κ) = 0.963). The Bland–Altman evaluation of the abnormal cell proportion between the AI analysis and manual analysis showed that the bias ± SD was 0.752 ± 6.646, and the 95% limit of agreement was from −12.775 to 13.779 (*p* = 0.1225). The total consistency of the AI immunophenotypic diagnosis and the manual results was 0.889 (kappa, 0.775). The consistency and speedup of the AI-assisted workflow indicate its promising clinical application.

## 1. Introduction

Leukemia is a malignant clonal disease of hematopoietic stem cells. The diagnosis of leukemia is made by a combination of clinical findings, morphologic examination of peripheral blood (PB) and bone marrow (BM) specimens, and cytogenetic and molecular data, as well as immunophenotypic analysis by multiparameter flow cytometry (MFC) [1,2,3]. In the last 30 years, MFC has become an essential tool in the diagnosis of leukemia [1,4]. Through the comprehensive assessment of the surface and intracellular antigens expressed by leukemic blasts, MFC enables pathologists to detect the blast lineage assignment and identify aberrant immunophenotypic features, allowing for the distinction of abnormal blast populations from normal progenitors. As the data interpretation is sophisticated, only skilled and highly trained pathologists are competent in it, which compromises the wide application of MFC. This is particularly so in countries like China, where the number of pathologists is insufficient, and the workload is large. More attempts have been made to solve the flow cytometry data analysis problem, and general-purpose dimension reduction and clustering algorithms including T-SNE and K-means have been used to address the analysis of MFC data [5,6]. Many FCM-specific algorithms, such as SPADE, FlowSOM, and PhenoGraph, have also been developed for MFC data processing [7,8]. However, clinical MFC data analysis still heavily depends on the manual logic gate strategy with conventional flow cytometry analysis software, and the detection efficiency is highly reliant on the examiners’ experience [9]. In addition, it is difficult to separate and set the gate for cells with high-dimensional data because the manual logic gating is limited to two-dimensional scattered plot combinations. Moreover, it is difficult to consistently measure the numerical expression levels of various antigens in the multidimensional space of the target cell group [10]. Therefore, there is a strong need for a flow cytometry analysis methodology that is exempt from the human factor and can simultaneously analyze multiple cell groups and their antigen expression levels in a multidimensional space so that leukemia cells can be detected more consistently and objectively. With the fast evolution of AI technology and its applications in medicine in recent years, this goal has become possible.

As reported, AI has been used in the prognosis of breast cancer and gastric cancer, as well as the diagnosis of colorectal cancer and the differential diagnosis of malignant or benign masses in the breast [11,12]. Moreover, the accuracy measure for the classification task has improved, owing to the use of the automatic analysis of hematoxylin- and eosin-stained histological images [13]. Therefore, AI-based clinical cancer research has resulted in a paradigm shift in cancer treatment. It is logical to expect that such advances of AI technology will help solve the challenges of cancer prognosis and diagnosis.

In this study, we evaluated an AI-assisted methodology for the diagnosis of leukemia, including the diagnostic results, abnormal cell proportion, and cell phenotypic diagnosis, as well as compared the method’s diagnostic accuracy with that of conventional manual analysis. A clinic-orientated AI-assisted diagnosis workflow was validated to perform automatic data analysis with not only the final diagnostic results, but also human-understandable and editable intermediate steps.

## 2. Materials and Methods

### 2.1. Study Groups

The study group consisted of 200 acute leukemia patients who were referred to the First Affiliated Hospital of Sun Yat-Sen University from March 2019 through June 2020. The diagnosis of acute leukemia was made according to the current WHO classification criteria by a combination of clinical findings, morphologic examination of PB and BM specimens, and cytogenetic and molecular data [14]. Cases with equivocal findings or insufficient data to establish disease, or that excluded diagnosis of acute leukemia, were excluded from this group. All of the patients had at least one diagnostic BM aspirate sample submitted for FCMat to the flow cytometry laboratory of the First Affiliated Hospital of Sun Yat-Sen University. The non-leukemic group included 94 patients with cytopenia(s) or hematocytosis attributable to a variety of non-neoplastic conditions. The clinicopathologic information was obtained by reviewing the medical records. No additional patient consent was obtained because it was a retrospective study. This study was approved by the Ethical Committee of the First Affiliated Hospital of Sun Yat-Sen University.

### 2.2. MFC Immunophenotyping

BM aspirate samples were collected in EDTA anticoagulant and processed within 24 h of collection. After incubation with monoclonal antibodies for 20 min at room temperature, erythrocytes were lysed with ammonium chloride (PharmLyse™, BD Biosciences, San Diego, CA, USA) at room temperature for 10 min using a standard lysing/washing technique. An eight-color FCM analysis was performed on a FACS Canto Plus flow cytometer (BD Biosciences, San Jose, CA, USA), which was standardized daily using CS&T beads, and data were analyzed. The panel of antibodies used was CD45 (for identifying blasts, was added in all of the tubes), CD2, CD3, CD4, CD5, CD7, CD8, CD10, CD11b, CD13, CD14, CD15, CD16, CD19, CD20, CD22, CD33, CD34, CD38, CD56, CD64, CD117, HDL-DR, MPO, CD79a, and cCD3. Some cases of AML, such as AML M6, CD235a and CD71, as well as AML M7, CD41, CD42b, and CD61, were also included in the panel. The antibody panel is shown in Appendix A. Instrument alignments, sensitivities, and spectral compensation were verified by standards, calibrators, procedural controls, and normal peripheral blood samples prior to processing of patient samples.

### 2.3. AI-Assisted Flow Cytometry Analysis Workflow

Deepflow is an AI-assisted flow cytometry analysis software that was developed by DeepCyto LLC, WestLinn, OR, USA. DeepFlowVeriosn 1.0.1 was evaluated in this study on both leukemia and non-leukemic cases. The AI-assisted workflow consists of five major analysis phases: (1) data validation, (2) population classification, (3) immune-phenotype classification, (4) AI-assisted diagnosis, and (5) report generation. These five phases are discussed in detail in below.

The data validation incorporates multiple machine learning models to extract nucleated single cells from the raw MFC data, such as flow time stability screening, doublets filtering, and debris removal, as shown in Figure 1. In the flow stability screening, the algorithm checks the moving average (also called the rolling average or running average) of the forward scattering signal and raises a warning message if there is an inconsistent change in data acquisition, which indicates a sudden change in the flow, voltage, or other experimental conditions. To filter out the doublets, a linear regression model obtains the coordinates of the linear separator for single cells versus doublets in FSC-A and FSC-H. As in the manual gating practice, the debris is characterized as the cell cluster at the lower side on FSC-A in the FSC-A/SSC-A plot, as shown in Figure 1. An unsupervised learning algorithm is used to cluster the cells, and a supervised learning model is used to identify the debris cluster based on its MFI on FSC-A and SSC-A. The machine learning models involved in the automatic data preparation were trained from 500 cases and validated in another 227 cases [15].

In the population classification phase, a multidimensional density–phenotype coupling (MDPC) algorithm specially developed for MFC data is applied on all of the nucleated cell data. The algorithm considers all of the channels’ distributions and phenotypes at the same time and automatically adjusts a cluster’s span according to the overall distribution. Two main criteria are used to differentiate the cell populations: the cell distribution density across all of the markers, and the marker expression phenotype on all of the markers (i.e., the MFI and relative expression level, respectively). The pervasive expression levels of each channel are classified into five levels: bright, positive, partial, dim, and negative. A new cell population will be created whenever these two criteria are met. At the end of this phase, all of the nucleated cells are clustered into cell groups, each of which is given a unique population ID. Considering that acute leukemia usually reaches above 20% in terms of the abnormal cell percentage, the MDPC algorithm is optimized for large cell groups (5% and above).

All of the cell clusters from the previous phase are classified in the population classification phase. In this phase, a random forest classifier with bootstrap aggregating is built for each tube. According to each tube’s specific antibody combinations, the AI models can not only characterize five common categories (i.e., lymphocytes, monocytes, granulocytes, blasts, and nucleated red blood cells (NRBCs)) in each tube, but they can also distinguish some subcategories, such as T-cells and B-cells, and identify blasts’ lineages. Appendix A shows the cell categories for each tube’s AI model alongside its antibody combination. A cell cluster’s raw cell attributes are encoded into a cluster-level feature vector including statistical parameters such as the MFI, standard deviation, and distributions of each channel. Thus far, the AI model is used for tube-wise classification; however, there are scenarios where the AI model cannot clearly categorize some of the cell clusters without the antibody expression information from other tubes. Therefore, a cross-tube match algorithm is employed to improve the accuracy of cell classification. For the target cell cluster in one tube, the algorithm searches all of the cell clusters in another tube and uses all of the shared channels to get the anchor features to find the best matched cell clusters. With cross-tube antibody expression patterns, the integrated AI model can identify most cell clusters.

In the AI-assisted diagnosis phase, a diagnosis AI model gathers all of the previously obtained information, affirms acute leukemia subtypes and negative cases, and raises diagnostic warnings. The AI model adopts a boosted random forest algorithm and takes the input of all of the categorized cell clusters, which are parameterized based on its ratio, category, and phenotypes. For the normal cell clusters, such as lymphocytes, monocytes, and granulocytes, the AI model examines and reports any unusual expression. The AI model also analyzes the abnormal cell clusters, combines clusters with similar expressions, computes the cell percentages, and diagnoses the acute leukemia’s subtype.

The report generation is the last phase of the AI-assisted workflow. Along with the diagnosis, 2D scatterplots that reveal the intermediate results (e.g., doublets/debris removal, cell clustering, and abnormity recognition) are automatically integrated into a FCM diagnosis report in one PDF file, as shown in Figure 1. The pathologist can easily evaluate and review each stage in the AI-assisted diagnostic workflow, and then decide to accept, adjust, or reject the AI results. In addition to the 2D scatterplots, more graphic plot options are also supported for visualizations of the cell population phenotype, including t-distributed stochastic neighbor embedding (t-SNE) and heat maps, as shown in Figure 1. The automatically generated FCM diagnosis report also includes quality control measures, inspired by real-world scenarios such as unexpected signal shifts, ratio abnormities, and flow instability. Moreover, quality-control indicators are checked at critical stages to ensure that the automatic workflow runs under normal conditions; otherwise, it raises an alert of possible manual interference.

### 2.4. Comparison of AI Results with Manual Results

The consistency analysis of AI results and manual results was carried out by kappa analysis and Bland–Altman analysis. SPSS software was used for kappa analysis, and Graphpad Prism 5 was used for Bland–Altman analysis.

### 2.5. Statistical Analysis

Statistical analysis was carried out by SPSS Statistics version 25 and Graphpad Prism 5. A two-tailed *p* value of less than 0.05 was considered to be statistically significant.

## 3. Results

### 3.1. Clinical Characteristics and Visualization of Results

The leukemia-positive group included 200 acute leukemia patients: 95 men and 105 women. The non-leukemic group included 44 men and 50 women. The non-leukemic group included five patients with aplastic anemia and 94 patients with cytopenia(s) or hematocytosis attributable to a variety of non-neoplastic conditions, including infection (*n* = 23), postchemotherapy bone marrow suppression (*n* = 26), autoimmune cytopenia(s) (*n* = 16), chronic renal insufficiency (*n* = 9), iron deficiency anemia (*n* = 6), and drug-induced cytopenia(s) (*n* = 9). The mean age of the leukemia-positive group and non-leukemic group was 43.12 ± 11.05 and 39.59 ± 9.18 (years), respectively. The sex and age between the leukemia patients and non-leukemic-patients had no statistically significant difference (*p* > 0.05).

The AI-assisted workflow is illustrated in Figure 2. The representative scatter diagram, heat map, and TSNE plot of leukemia patients and non-leukemic cases are shown in Figure 3 and Figure 4.

### 3.2. Comparison of Diagnostic Results

First, we compared the consistency of the diagnostic results by AI and manual results. The consistencies of the non-leukemic, AML, B-ALL, and T-ALL cases were 1, 0.971, 0.981, and 0.778, respectively. Moreover, the total consistency was 0.976 (Table 1). The kappa (K) value was 0.963, which indicates that the diagnostic results of AI had good consistency with that of the manual results. Further, the AI analysis time (83.72 ± 23.90 s, mean ± SD) was distinctly shorter than the mean time in the manual results (15.64 ± 7.16 min, mean ± SD).

### 3.3. Comparison of Abnormal Cell Proportion

The abnormal cell proportions of the AI and manual results were 64.487 ± 23.36 and 62.973 ± 22.693, respectively. As shown in Table 1, the Pearson correlation coefficient was 0.913 (*p* < 0.04), which indicates that the abnormal cell proportion of AI and manual results had significant correlation. As shown in Figure 5, Bland-Altman was used to evaluate the abnormal cell proportion between the AI analysis and manual analysis results. The bias ± SD was 0.752 ± 6.646, and the 95% limit of agreement was from −12.775 to 13.779. The paired t test showed that the abnormal cell proportion between AI analysis and manual analysis was not significantly different (*p* = 0.1225).

### 3.4. Comparison of AI Immunophenotypic Diagnosis and Manual Results

As shown in Table 2, the antigenic feature of the neoplastic cells and the expression levels were used in this comparison. The total consistency of the AI cell phenotypic diagnosis and manual results was 0.889 (kappa, 0.775), which indicates that the AI cell immunophenotypic diagnosis and manual results had good consistency.

## 4. Discussion

MFC plays a crucial role in the diagnosis of acute leukemia. In the past decade, many machine learning attempts have made been to solve the flow cytometry data analysis problem. Most machine learning applications in MFC data focus on the biomedical research field, but an AI solution for clinical flow cytometry application is still lacking for several reasons.

First, the motivation of the research flow cytometry is to be innovative, which pushes the continuous trial of different panel designs with constant changes in reagents in a small batch of samples [16,17]. In contrast, in the clinical practice of flow cytometry, the priority is safety and efficacy [18,19], so many clinical flow cytometry labs perform MFC tests with a focus on stable panels with proven records and a fast turnaround time. Additionally, research-driven flow cytometry is funded by research grants, so time and cost factors are not as critical as those in the clinical world, where turnaround time and economics are the lab manager’s highest priorities. Many clustering algorithms like T-SNE have flourished in research, but are seldom used in clinical scenarios owing to the slow performance caused by the iterative nature of the t-SNE algorithm [20,21]. It takes hours or even days to do T-SNE with million-event MRD MFC data, which is usually unacceptable in clinical practice [22].

Second, most published AI attempts in the flow cytometry field have been based on an end-to-end black-box approach, which lack the human understandable intermediate diagnosis results, and clinical practitioners have found them to be difficult to review and validate [23,24]. Therefore, there is still a strong need for an AI flow cytometry solution with human-explainable results and intermediate steps.

In our study, we established an AI model to analyze the data of MFC, which can assist examiners in making a diagnosis of leukemia. Our results show that the AI model could quickly detect abnormal cell proportion and cell phenotype to obtain the diagnostic results. Comparably, the consistency of diagnostic results between AI analysis and manual analysis was high. Seven cases were manually diagnosed as leukemia (four AML and two T-ALL, including one early T-cell precursor leukemia and one B-ALL). In contrast, AI only found abnormal blasts and raised a prompt for manual review. These cases were not easily diagnosable cases. All four AML cases were MPO negative and cross-expressed lymphoid antigens such as CD7 or/and CD56. Moreover, in the three ALL cases, the expression of the lymphoid-lineage-specific marker such as cCD3 or CD79a was dim, and myeloid antigens such as CD13 or/and CD33 were cross-expressed. Actually, the correct diagnosis of such cases is challenging for tyro. In this situation, AI did give abnormal-diagnosis hints, indicating that further manual review was needed. Of course, in the future, with a larger training set containing more atypical phenotypes, AI can be used to make a more definitive diagnosis, even for less common cases. Moreover, the detection of abnormal cell percentage and MFI results were also in close numerical range.

The average time of the manual analysis of each sample FCM assay test was about 15 min by experienced examiners; the analysis speed of the AI model (83.72 ± 23.90 s, mean ± SD) was 10 times faster than that of traditional manual analysis. In addition, AI analysis significantly reduced the error of entry and calculation introduced by manual analysis.

As shown in Appendix A, the difference of the abnormal cell proportion between AI analysis and manual analysis beyond 20% was 12 patients, including nine more manual analysis patients and three more AI analysis patients. The cause of the higher AI proportion may be that AI misdiagnosed some granulocytes/monocytes as abnormal cells. On the contrary, the cause of a higher manual analysis proportion may be that AI was too sensitive to some unclassified cells that needed to be analyzed manually. However, the Pearson correlation coefficient of cell proportion was 0.913 (*p* < 0.04), which indicates a good correlation of the AI model and manual analysis.

As shown in Table 2, the immunophenotypic diagnostic consistency between AI and manual analysis ranged from 0.75–0.99 (Kappa (K) range: 0.139–0.925). The total consistency of AI cell phenotypic diagnosis and manual results was 0.889 (kappa, 0.775), which indicates that AI cell phenotypic diagnosis and manual results had good consistency. Moreover, the final diagnostic results between the AI model and manual analysis had good consistency (total consistency: 0.976; kappa (K): 0.963).

We investigated the potential for using the AI-assisted methodology to help pathologists identify abnormal cell populations faster and more objectively. This study focused on acute leukemia cases. More flow cytometry panels (including B-ALL MRD, AML MRD, and B-Cell Lymphoma) will be included in future research to test the generalization ability of the AI methodology for flow cytometry. Further validation will be made by testing the AI methodology on the variant FCM panels from different laboratories. As flow cytometry is highly customized in panel design, AI can help standardize the hematology diagnosis in flow cytometry. The AI model established in this study is mainly based on clustering, classification, and dimension-reduction algorithms. In the future, we will try to use the method of a convolutional neural network [25] to more effectively detect acute leukemia cells, reduce false positives, and improve the accuracy of AI-assisted diagnosis of acute leukemia.

In summary, a clinic-orientated AI-assisted diagnosis workflow was validated to perform automatic data analysis with not only the final diagnostic results, but also human-understandable and editable intermediate steps. Specifically, the AI-assisted diagnostic system adopts customized machine learning algorithms for all of the clinical MFC analysis stages, and each step produces intermediate human-readable results that follow the same diagnosis logic in the manual analysis workflow. Therefore, a pathologist can easily understand and evaluate the AI-generated results. Moreover, the AI-assisted workflow supports interactive editing in each step, where the pathologist can decide to accept or adjust the results. This provides extra flexibility so that AI can adaptively learn each pathologist’s personal preferences in the gating and diagnosis process. AI-assisted automated MFC analysis is promising in leukemia diagnosis, and it is more rapid and effective. In addition, it can be integrated with other test findings, such as morphological, cytogenetic, and molecular abnormalities, to diagnose and stratify the prognosis of acute leukemia. Although its clinical application still needs to be further validated, this scalable system can be used as the basis for a clinical decision-making support system.

## Figures and Tables

**Figure 1 diagnostics-12-00827-f001:**
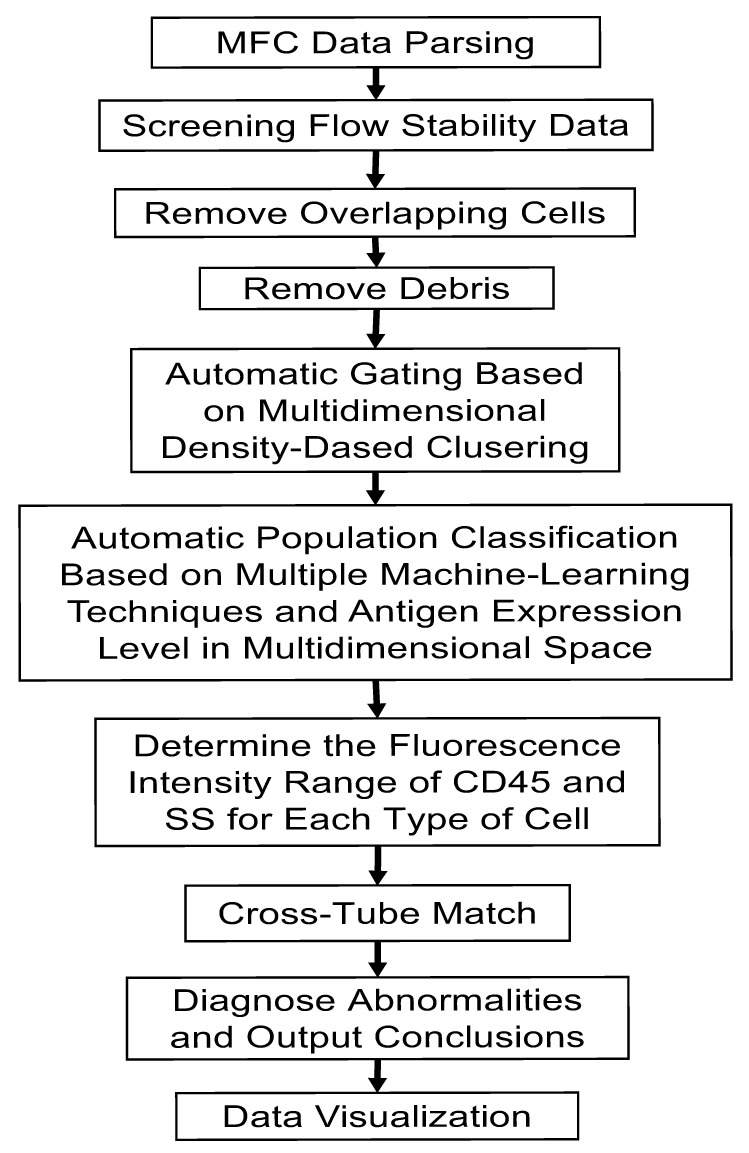
Workflow of MFC results.

**Figure 2 diagnostics-12-00827-f002:**
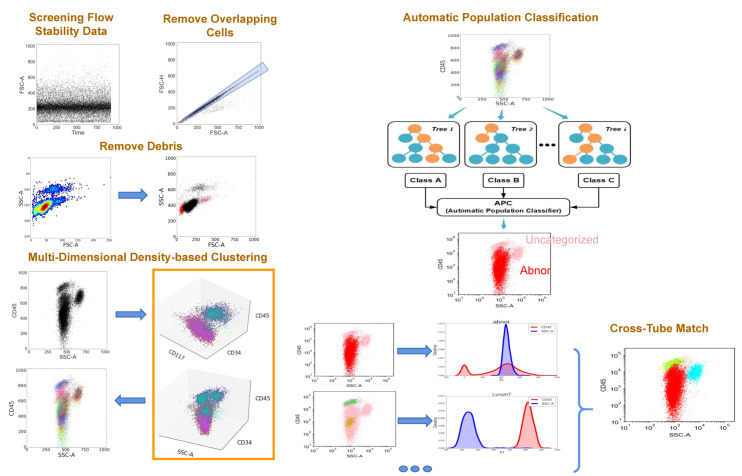
Output and visualization of the results.

**Figure 3 diagnostics-12-00827-f003:**
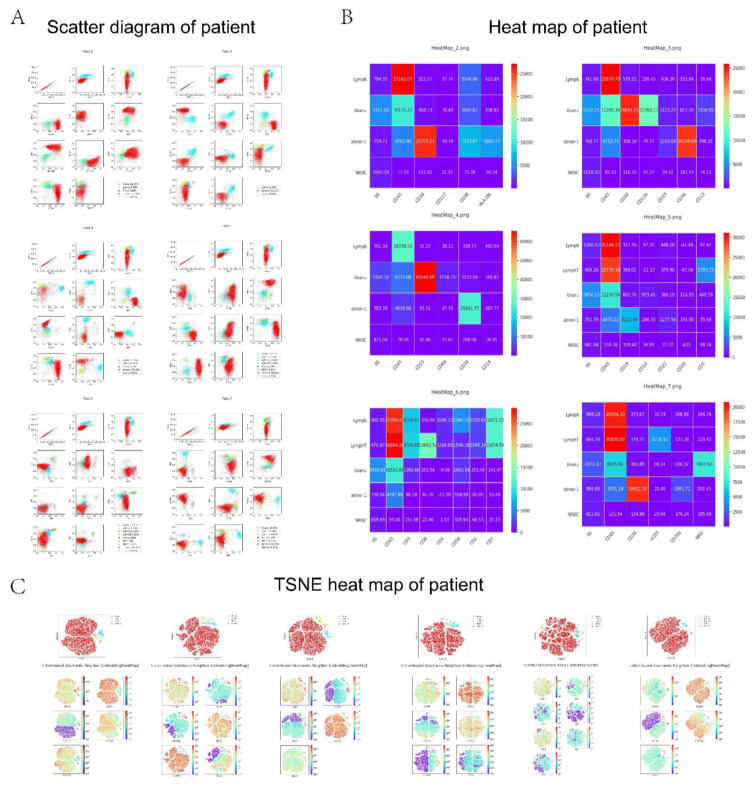
The representative scatter diagram, heat map, and TSNE plot of the patient. (**A**) Scatter diagram; (**B**) heat map; and (**C**) TSNE heat map.

**Figure 4 diagnostics-12-00827-f004:**
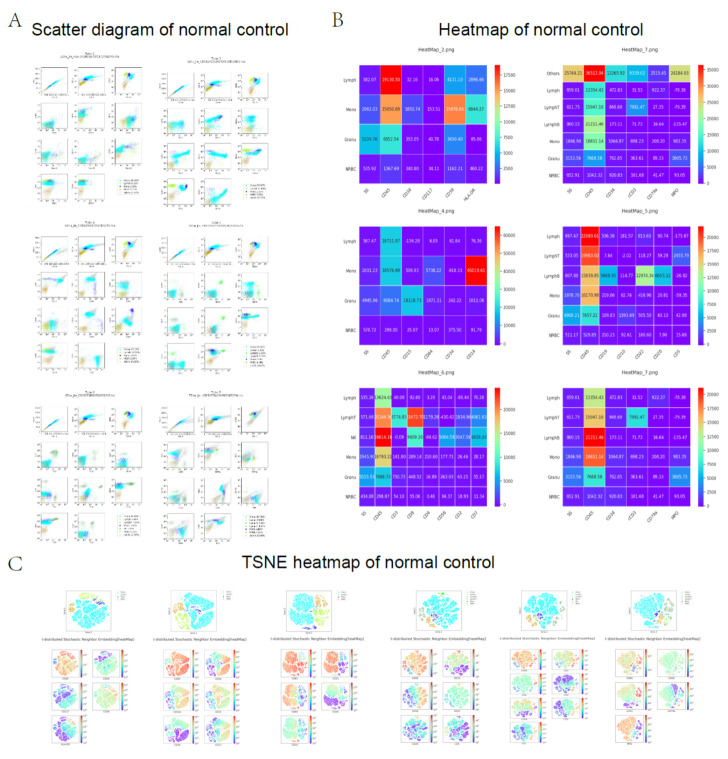
The representative scatter diagram, heat map, and TSNE plot of normal control. (**A**) Scatter diagram; (**B**) heat map; and (**C**) TSNE heat map.

**Figure 5 diagnostics-12-00827-f005:**
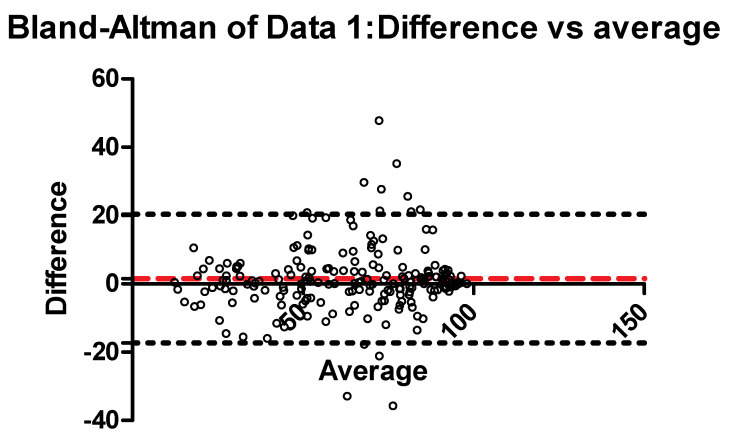
The evaluation of the abnormal cell proportion. The bias ± SD was 0.752 ± 6.646, and the 95% limit of agreement was from −12.775 to 13.779.

**Table 1 diagnostics-12-00827-t001:** Comparison of diagnostic results.

	AI	AML	B-ALL	Normal	T-ALL	Abnormal	Total	Consistency
Manual								
AML		134	0	0	0	4	138	0.971
B-ALL		0	52	0	0	1	53	0.981
T-ALL		0	0	0	7	2	9	0.778
Normal		0	0	94	0	0	94	1.000
Total		134	52	94	7	7	294	0.976
		abnor-1_manual				abnor-1_AI		
abnor-1_manual		1				0.913 **		
abnor-1_AI		0.913 **				1		

**: *p* < 0.01.

**Table 2 diagnostics-12-00827-t002:** Comparison of AI cell phenotypic diagnosis and artificial results.

Manual- AI	Pos- Pos	Pos- Neg	Pos- Partial	Partial- Pos	Partial- Partial	Partial -Neg	Neg- Pos	Neg- Partial	Neg- Neg	Total	Consistency	Kappa (K)
**HLD-DR**	115	9	0	11	21	1	0	4	39	200	0.875	0.768
**CD117**	61	7	0	22	35	3	0	6	66	200	0.81	0.713
**CD34**	90	3	0	10	40	2	0	5	50	200	0.9	0.842
**CD38**	135	35	0	7	18	2	0	1	2	200	0.775	0.375
**CD16**	0	0	0	0	3	1	0	1	195	200	0.99	0.745
**CD11b**	2	1	0	2	10	17	0	3	165	200	0.885	0.489
**CD13**	63	3	0	25	51	2	0	20	36	200	0.75	0.612
**CD33**	84	4	0	22	40	6	0	7	37	200	0.805	0.692
**CD15**	5	4	0	2	20	25	0	3	141	200	0.83	0.539
**CD64**	15	3	0	12	26	9	0	12	123	200	0.82	0.636
**CD14**	0	0	0	1	0	5	0	0	194	200	0.97	0.139
**CD5**	3	2	0	0	1	4	0	2	188	200	0.96	0.54
**CD10**	43	0	0	3	4	0	0	3	147	200	0.97	0.925
**CD22**	31	6	0	2	14	2	0	8	137	200	0.91	0.801
**CD20**	10	3	0	3	13	6	0	5	160	200	0.915	0.716
**CD19**	49	1	0	1	10	0	0	20	119	200	0.89	0.787
**CD7**	17	11	0	3	18	4	0	3	144	200	0.895	0.751
**CD2**	4	2	0	0	2	6	0	1	185	200	0.955	0.592
**CD56**	6	1	0	3	16	8	0	1	165	200	0.935	0.758
**CD3**	0	1	0	0	2	2	0	0	195	200	0.985	0.619
**CD4**	6	1	0	4	13	23	0	12	141	200	0.8	0.42
**CD8**	3	0	0	0	7	3	0	2	185	200	0.975	0.789
**MPO**	46	2	0	17	28	7	0	12	88	200	0.81	0.7
**CD79a**	28	9	0	3	13	1	0	19	127	200	0.84	0.671
**cCD3**	3	3	0	0	2	1	0	1	190	200	0.975	0.713
**Total**	819	111	0	153	407	140	0	151	3219	5000	0.889	0.775

## Data Availability

The data and material in our studies are available upon request to the corresponding author.

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
