# Peer review of "Diagnosis of Acute Leukemia by Multiparameter Flow Cytometry with the Assistance of Artificial Intelligence"

_diagnostics, 2022, doi:10.3390/diagnostics12040827_

Round 1
Reviewer 1 Report
-
In this paper, the Authors evaluate a novel machine learning based method of AI assisted diagnosis of acute myeloid leukemia with multicolour flow cytometry. MFC is an important diagnostic tool able to identify different bone marrow and peripheral blood cell populations by analysing surface and cytoplasmatic antigen expression. Specifically, MFC is the most widely used tool in acute leukemia diagnosis and it is able to precisely define the lineage and its use at diagnosis is recommended by international guidelines such as European Leukemia Net. MFC is also widely used during AML treatment as a tool for minimal residual disease assessment.
However, MFC analysis in AML requires a skilled biologist/pathologist as the interpretation of the results, especially in the MRD setting.
This may be difficult in lower income countries and in this view AI and machine learning may significantly help.
In this paper, the Authors use the software Deepflow v1.0.1 and they evaluate its performance in analysing BM samples both from AML patients and from patients with non-oncological diseases. The AI-assisted workflow consisted of 5 major analysis phases: validation, population classification, IF classification, AI assisted diagnosis and final report.
The software showed good correlation with manual analysis and employed a significantly lower time.Overall, the research is well conduct and results are convincing.
Main Issues
- The paper should be thoroughly checked by a native English speaker, as many sentences are confusing
Author Response
Response to Reviewer 1 Comments
Point 1: The paper should be thoroughly checked by a native English speaker, as many sentences are confusing
Response: We have sought professional help to improve the language. The revised manuscript and layout has been edited and proofread by CM-H of Let-Pub.

Reviewer 2 Report
Diagnostics Review
Manuscript ID: diagnostics-1625852
Type of manuscript: Article
Title: The diagnosis of acute leukemia by multiparameter flow cytometry with
assistance of artificial intelligence
Authors: Pengqiang Zhong, Mengzhi Hong, Huanyu He, Jiang Zhang, Yaoming Chen,
Zhigang Wang, Peisong Chen *, Juan Ouyang *
Submitted to section: Pathology and Molecular Diagnostics,
Summary:
Zhong P et al. introduced artificial intelligence (AI) technology to multiparameter flow cytometry (MFC) analysis for acute leukemia diagnosis and compared it with manual diagnosis by a pathologist to see if it can accurately diagnose acute leukemia. The results showed a high consistency rate between AI and human manual diagnoses, with comparable diagnostic accuracy. Furthermore, it became apparent that diagnosis with AI was faster than manual diagnosis and may allow to analysis of a larger number of samples. These results are very interesting and deserve to be reported as there are few similar reports. However, the following points need to be corrected and added.
Major point:
According to instruction for authors, Figures and Schemes to be provided must be of sufficient resolution (minimum 1000 pixels width/height, or a resolution of 300 dpi or higher); however, Figures 2, 3, and 4 are of low resolution and their contents are unclear. In particular, Figures 3 and 4 are completely illegible and cannot be reviewed. Proofreading, including layout, is needed.
Minor points:
Materials and Methods part
#1 Although "the control group" is mentioned, the name "control group" should be avoided because this is not a normal control but a non-leukemic group with aplastic anemia and cytopenia/hyperleukocytosis due to various causes.
#2 Has the AI technology analysis software (Deepflow) employed in this study been used in similar clinical studies before? This machine learning model states that the model was trained on 500 cases and validated on 225 cases, but is this number of cases sufficient?  In addition, the name of the authors of the reference [15] related to this description does not match the instructions for authors and the article could not be found in Pubmed data base.
#3 2.4 Comparison of AI "Eesults" with Manual Results. "Eesults" is considered a typo in “Results”.
Results part
#4 In the present study, a large number of panels were used with MFC. In addition, 134 of the 200 patients had AML.Among AML, FAB classification could be diagnosed by AI technology?
Discussion part
#5 Table 1 shows the consistency rate of leukemia diagnosis between AI and manual. Discussion is needed on the causes of the discrepancy in leukemia diagnosis and what efforts are needed to further improve the consistency rate.
Author Response
Response to Reviewer 2 Comments
Point 1: According to instruction for authors, Figures and Schemes to be provided must be of sufficient resolution (minimum 1000 pixels width/height, or a resolution of 300 dpi or higher); however, Figures 2, 3, and 4 are of low resolution and their contents are unclear. In particular, Figures 3 and 4 are completely illegible and cannot be reviewed. Proofreading, including layout, is needed.
Response 1: Thank you for your suggestion. We have provided the figures and schemes with sufficient resolution. The layout has been proofread too.
Point 2: Although "the control group" is mentioned, the name "control group" should be avoided because this is not a normal control but a non-leukemic group with aplastic anemia and cytopenia/hyperleukocytosis due to various causes.
Response 2: We agree with you on that point and made changes accordingly. We have replaced “control group” with “non-leukemic group” in the paper.
Point 3:Has the AI technology analysis software (Deepflow) employed in this study been used in similar clinical studies before? This machine learning model states that the model was trained on 500 cases and validated on 225 cases, but is this number of cases sufficient?  In addition, the name of the authors of the reference [15] related to this description does not match the instructions for authors and the article could not be found in Pubmed data base.
Response 3: DeepFlow has been used in multiple clinical studies for different FCM panels, including BALL -MRD panel at the first hospital of Peking University(Published in Dec. 2020), CLL-MRD and BALL-MRD at MD Anderson Cancer Center, Houston(To be published), ALPS immunology monitoring FCM panel at Oregon Health Science University, Portland, Oregon (accepted at 2022 AACC conference).
For machine learning application with structured data like flowcytometry, usually for each AI target category (in this study, about 10 cell population categories including, Blast, Lymphocyte, Monocyte, Myeloids…) a rule of thumb is that each category need 1000 sample data for training to achieve satisfying AI prediction accuracy. For this flowcytometry, each tube has more than 30,000 events, and the single test case of 7 tubes has more than 200,000 data points on average. For 500 cases, the total training data amount is more than 100 million data points, so for each category of the cell population we are training, there are more than 10 million training data points, which is way more than industry standard(1000 per category), even with considering of the uneven distribution of different cell type.
We have revised the format of reference [15] according to the instructions for authors. And we will provide the article in attachments.
Point 4: 2.4 Comparison of AI "Eesults" with Manual Results. "Eesults" is considered a typo in “Results”.
Response 4: Yes, it is a typo. Thank you for your careful review.
Point 5:In the present study, a large number of panels were used with MFC. In addition, 134 of the 200 patients had AML. Among AML, FAB classification could be diagnosed by AI technology?
Response 5: You raised a good question. In our country, WHO classification is more often used in the diagnosis of acute leukemia. FAB classification of leukemia is a morphological classification, which is also widely used. At present, our AI technology cannot give FAB classification results, but with the accumulation of data, it would be an interesting attempt in the future.
Point 6: Table 1 shows the consistency rate of leukemia diagnosis between AI and manual. Discussion is needed on the causes of the discrepancy in leukemia diagnosis and what efforts are needed to further improve the consistency rate.
Response 6: We agree that we should discuss more about the causes of the discrepancy. In our study, 7 cases were manually diagnosed as leukemia (4 AML, 2 T-ALL including 1 early T-cell precursor leukemia, 1 B-ALL); while AI only found abnormal blasts and prompted manual review. These 7 cases are not easy cases. All 4 AML cases were MPO negative and cross-expressed lymphoid antigens such as CD7 or/and CD56; in 3 ALL cases, the expression of lymphoid lineage specific marker such as cCD3, CD79a was dim, and myeloid antigens such as CD13 or/and CD33 were cross-expressed. Actually, the correct diagnosis of such cases is challenging for tyro. In this situation, AI did give abnormal hints that further manual review is needed. Of course, in the future, with bigger training set containing more atypical phenotypes, AI can give more affirmative diagnosis even for less common cases. We’ve also made relevant changes in the paper.